# Genome-Wide Analysis of the Polygalacturonase Gene Family in Macadamia and Identification of Members Involved in Fruit Abscission

**DOI:** 10.3390/plants14111610

**Published:** 2025-05-25

**Authors:** Yu-Chong Fei, Yi Mo, Jiajing Xu, Kai Lin, Liang Tao, Xiyong He, Meng Li, Zeng-Fu Xu

**Affiliations:** 1Guangxi Key Laboratory of Forest Ecology and Conservation, State Key Laboratory for Conservation and Utilization of Subtropical Agro-Bioresources, College of Forestry, Guangxi University, Nanning 530004, China; feiyc@st.gxu.edu.cn (Y.-C.F.); 2109401003@st.gxu.edu.cn (Y.M.); 2009392033@st.gxu.edu.cn (J.X.); linkai@st.gxu.edu.cn (K.L.); 15935412298@163.com (M.L.); 2Guangxi Colleges and Universities Key Laboratory for Cultivation and Utilization of Subtropical Forest Plantation, Key Laboratory of National Forestry and Grassland Administration on Cultivation of Fast-Growing Timber in Central South China, College of Forestry, Guangxi University, Nanning 530004, China; 3Yunnan Institute of Tropical Crops, Jinghong 666100, China; basanyeyu@vip.sina.com (L.T.); heda0691@163.com (X.H.)

**Keywords:** macadamia, PG gene family, fruit abscission, expression profile, transient overexpression

## Abstract

Severe physiological fruit abscission significantly limits yield potential in macadamia. Polygalacturonase (PG), a key hydrolytic enzyme in pectin degradation, plays a critical role in fruit abscission. However, in the macadamia genome, the *PG* gene family and the members involved in fruit abscission remain poorly understood. In this study, 56 *PG* gene family members, which were unevenly distributed across 13 of the 14 chromosomes, were identified in the macadamia genome. Phylogenetic analysis clustered these genes into seven clades, with most members found in clades D and E. The *MiPGs* contained 3–11 exons and 2–10 introns, and except for those in clades E and G, most contained conserved domains I–IV and were predicted to be localized exclusively to the cell membrane. *MiPG* promoter analysis revealed numerous light-, phytohormone-, and stress-responsive *cis*-elements. Expression profiling during fruit development showed that twelve *MiPGs* were either undetectable or expressed at low levels in the fruit abscission zone, whereas eight were highly expressed. *MiPG9*, *MiPG37*, and *MiPG53* were significantly upregulated during abscission induced by a combination of girdling with defoliation and ethephon treatments. Moreover, transient *MiPG37* overexpression in lily petals promoted premature abscission, suggesting that this gene plays a pivotal role in macadamia fruit abscission. These findings advance the functional characterization of macadamia *PG* genes and highlight a subset of candidate genes for further genetic manipulation to improve fruit retention.

## 1. Introduction

Abscission is a developmentally controlled program for cell separation [1] in which plant organs (e.g., leaves, flowers, and fruit) are no longer conducive to the survival of the parent plant or as a step of reproductive development [2]. For fruit to abscise, cell separation must occur in a precise location called the abscission zone (AZ) [3]. Abscission occurs at the same time as the breakdown of the wall matrix that provides structure to the cells and tissues within the AZ, which is the result of cell wall polymer degradation and/or remodeling [3,4]. The plant cell wall is a complex, reticulate structure composed of cellulose, hemicellulose, pectin, and structural proteins [5,6]. Among these macromolecules, pectin is the major component of the middle lamella and primary cell wall and serves as a “cementing agent” to link cells [7,8]. Therefore, regulated pectin breakdown is essential for plant organ abscission. Pectin degradation involves coordinated actions of multiple enzyme classes, including polygalacturonase (PG), pectin lyase, pectin methylesterase, and β-galactosidase [3,4]. *PG* genes belong to one of the largest hydrolase families in plants [9] and play important roles in plant organ abscission [10,11]. The abscission of flowers, fruits, and leaves coincides with an increase in PG activity [10,12,13]. The expression levels of *PG* genes increase rapidly prior to plant organ abscission [14,15]. Transgenic studies on apple and tomato have confirmed the importance of *PG* genes in plant organ abscission. *PG* overexpression in apples results in premature leaf abscission due to reduced cell adhesion in the leaf AZ [16]. In contrast, virus-induced gene silencing has been used to silence *PG* genes in tomatoes, delaying abscission and increasing the break strength of the AZ in ethylene-treated explants [17]. Furthermore, 1-methylcyclopropene, a competitive inhibitor of ethylene activity, prevents cell separation in tomatoes by suppressing *PG* gene expression [18]. Consequently, increased PG activities and gene expression levels are common during plant organ abscission.

*PG* genes are involved in multiple plant developmental processes, including floral morphogenesis, fruit ripening, senescence, and organ abscission [6,9,19,20]. To date, *PG* gene families have been identified among plant species, including Arabidopsis [19], citrus [21], maize [22], peach [23], and tomato [24]. In Arabidopsis, the expression of 66 *PG* genes was detected in five tissues, with 40 *PG* genes in flowers, 34 in siliques and roots, 30 in leaves, and 31 in stems, but 23 *PG* genes had no detectable expression [9]. Research has shown that *ADPG1* and *ADPG2* are essential for silique dehiscence and that *ADPG2* and *QUARTET2* (*QRT2*) mediate floral organ abscission [20]. In Populus, two *PG* genes are specifically expressed in the leaf AZ under salt stress, indicating an association with leaf abscission [25]. In peaches, *PpPGs* are rapidly induced by ethylene to promote fruit softening [23]. In pear, the expression profiles of *PbrPGs* among tissues are distinct, with the highest expression levels in the stigma and the lowest in the petals [26]. *PG* genes are generally divided into six clades (A–F) based on their function and sequence characteristics. Clade A comprises genes expressed in the fruit and/or AZ that are related to fruit ripening and abscission [22,27,28,29]. In litchi, *LcPG1* (clade E) has been identified as a key regulator of fruit abscission [30]. These studies demonstrate that *PG* gene members exhibit tissue-specific and treatment-responsive expression patterns, highlighting functional diversification within this gene family.

Macadamia (*Macadamia integrifolia* and *M*. *tetraphylla*) is an evergreen tree native to rainforests in eastern Australia [31]. It is widely cultivated in tropical and subtropical regions of the world for its nutritious and delicious kernels [32]. During the flowering season, mature macadamia trees typically yield approximately 2500 racemes, each containing 100–300 flowers [33]. However, fewer than 10% of these flowers successfully set fruit at 2 weeks after anthesis (WAA), and more than 80% of the initial fruit is abscised at 3–8 WAA [33]. Severe fruit abscission is a primary constraint of macadamia yield. Therefore, reducing physiological fruit abscission and increasing yield have become important issues in macadamia-growing countries. A few studies have investigated abnormal fruit abscission in macadamia, evaluating cultural management techniques in the field [34,35,36], sink-source balance [37,38], and endogenous plant hormone concentrations [39], but there are insufficient reports on the molecular mechanism of macadamia fruit abscission. Furthermore, macadamia fruit is typically harvested from the orchard floor by mechanical sweepers after natural fruit abscission. Due to the difficulty of natural abscission, the prolonged harvest period, and the inconsistency of abscission times among cultivars, a considerable amount of human capital and financial resources are required for harvesting, particularly in mountainous regions, where mechanization is not readily available. Therefore, the synchronized abscission of macadamia fruit at full physiological maturity results in a significant reduction in the harvesting cost.

Despite the confirmation of the important role of *PG* genes in fruit abscission by numerous studies, there have been no studies related to *PG* gene family members in macadamia. Therefore, to identify which *PG* members are involved in macadamia fruit abscission, *PG* genes in macadamia were identified in this study via whole-genome retrieval and bioinformatics methods. Furthermore, the dynamic expression profile was analyzed, and several *PG* gene family members that may be related to fruit abscission were identified. The function of *MiPG37* was verified by transient overexpression in lily petals. These findings provide new insights into the role of *PG* genes in the abscission process of macadamia fruit.

## 2. Results

### 2.1. Identification of PG Gene Family Members in Macadamia

In total, 56 *PG* genes, which were named *MiPG1*–*MiPG56* according to their chromosomal locations, were identified from the *Macadamia integrifolia* HAES 741 genome [40] (Table 1 and Appendix A). Most *MiPGs* were unevenly distributed on 13 of the 14 chromosomes in the macadamia genome. *MiPGs* were identified, with one on chromosomes 9 and 10, two on chromosomes 1, 4, 6, and 7, three on chromosomes 13 and 14, four on chromosome 5, five on chromosome 12, six on chromosomes 2 and 3, seven on chromosome 11, twelve on unplaced scaffolds, and none on chromosome 8 (Appendix A).

Candidate *PG* gene family members containing at least two of the highly conserved *PG* domains (domains I (“SPNTDGI”), II (“GDDC”), III (“CGPGHGISIGSLG”), and IV (“RIK”)) were considered macadamia *PG* gene family members [41]. Most *PG* gene family members (36 members) contained conserved domains I, II, III, and IV, except for *MiPG22*, -*27*, -*28*, -*29*, and -*30* (the closest ortholog of *AtQRT3* (GenBank accession: AT4G20050)), which lacked the typical PG domain. *MiPG1*, -*8*, and -*18* lacked domain I, and *MiPG38* lacked domain II. *MiPG1*, -*2*, -*8*, -*14*, -*33*, -*35*, -*36*, -*37*, -*38*, -*40*, -*42*, -*49*, -*50*, and -*56* lacked domain III, and *MiPG56* lacked domain IV.

The encoded proteins ranged from 201 (*MiPG18*) to 519 (*MiPG9*) amino acids, with a molecular weight of 21.08–56.38 kDa. The isoelectric points ranged from 4.85 (*MiPG18* and *MiPG16*) to 9.45 (*MiPG27*), and the instability index, aliphatic index, and grand average of hydropathicity of *MiPGs* were within the ranges of 24.20–51.75, 73.34–99.80, and −0.296–0.071, respectively (Table 1 and Appendix A). The instability indices of most *MiPGs* were less than 40, suggesting their stability. The grand averages of hydropathicity for most *MiPGs* were less than 0, indicating that they are hydrophilic. Furthermore, signal peptides were predicted in 37 of the 56 *MiPG* members.

Except for *MiPG22* (cell membrane, chloroplast, and cytoplasm), *MiPG27* and *MiPG28* (chloroplast), and *MiPG30* (cell membrane and chloroplast), the subcellular localization of most *MiPGs* was predicted in the cell membrane (Table 1), which is consistent with the functions of these PG proteins. Variations in the structures and properties of *MiPGs* indicate that they have multiple functions in macadamia.

### 2.2. Phylogenetic Analysis of PG Gene Family Genes in Macadamia

A phylogenetic tree was constructed using the full-length protein sequences of 56 *MiPGs*, 66 *PG* genes from *Arabidopsis thaliana* (*AtPG*), and 26 *PG* genes from other horticultural plants with known fruit development-related functions, particularly those involved in abscission (Figure 1). The 138 *PG* genes were clustered into seven clades (A–G) based on previous studies [22,24,42]. Clade D had the most *PG* genes, and clade G had the least. Except for clade G, the number of *AtPGs* and *MiPGs* in the other clades was approximately equivalent. Clades A, B, C, D, E, F, and G contained 4, 4, 8, 12, 13, 10, and 5 *MiPGs*, respectively. Clade G was composed of *AtQRT3* and its closest orthologs (*MiPG22*, -*27*, -*28*, -*29*, and -*30*). Clade F consisted of only *MiPGs* and *AtPGs*, and clade C contained the most *PG* genes from horticultural plants. Five pairs of *MiPGs*, namely *MiPG4*–*MiPG5*, *MiPG12*–*MiPG13*, *MiPG24*–*MiPG41*, *MiPG24*–*MiPG47*, and *MiPG32*–*MiPG34*, had high degrees of homology in the terminal nodes, indicating that they are putative paralogous genes in the macadamia genome.

### 2.3. Gene Structure Analysis of PG Gene Family Genes in Macadamia

The molecular evolution of the *PG* gene family was primarily determined by the evolution *of* increasingly complex organs in plants [25]. A phylogenetic tree constructed from 56 *MiPG* protein sequences was consistent with the phylogenetic tree constructed from *PG* genes of macadamia and other species (Figure 2), which were also clustered into seven clades (clades A–G). To further analyze the conserved motifs in the amino acid sequences of *MiPGs*, 56 *MiPG* protein sequences were aligned using the online tool MEME to output eight conserved motifs, among which motifs 1, 4, 5, and 7 corresponded exactly to the four conserved domains of PG proteins (Appendix A). Within the same clade, the composition and positional order of these conserved motifs in the protein sequences of the *MiPGs* were similar. The *MiPGs* in both clades A and D contained seven motifs other than motif 8. The *MiPGs* of clade E lacked motifs 6 and 7 but contained motif 8. The closest homologs of *AtQRT3*, *MiPG27*, *MiPG29*, and *MiPG30* contained only motif 3; *MiPG22* contained motifs 3 and 4; and *MiPG28* did not contain any motifs.

The exon/intron structures and intron phases of *MiPGs* were analyzed using TBtools-Ⅱ v2.225, and their full-length coding sequences and corresponding genomic DNA sequences were used. The results revealed that the *MiPGs* consisted of 3–11 exons and 2–10 introns. The *MiPGs* of clades A and F contained more exons and introns than the *MiPGs* of other clades.

### 2.4. Cis-Element Analysis of the MiPG Genes

As *cis*-elements are important in the regulation of gene expression, we analyzed the *cis*-elements in the *MiPG* promoters using Plant CARE [43] (Figure 3). The promoters of the *MiPG* gene family contained 12 classes of abiotic stress *cis*-acting elements: abscisic acid (ABA) responsiveness, anaerobic induction, auxin responsiveness, defense and stress responsiveness, drought-inducibility, gibberellin responsiveness, light responsiveness, low-temperature responsiveness, methyl jasmonate responsiveness, salicylic acid responsiveness, wound responsiveness, and zein metabolism regulation. The number of *cis*-acting elements in the *MiPG* promoter sequences ranged from 3 (*MiPG20*) to 36 (*MiPG6*).

The most *cis*-acting elements in the *MiPG* promoters were associated with light responsiveness, followed by anaerobic induction and ABA responsiveness, and the least were associated with wound responsiveness. Light-responsive elements were detected in the promoters of all *MiPGs* and wound-responsive elements were only present in *MiPG6*, -*13*, -*16*, -*17*, -*21*, and -*47*.

### 2.5. Expression Profiles of PG Gene Family Genes in Macadamia

During macadamia fruit development, approximately 90.95% of the immature fruit underwent abscission (Appendix A). Fruit abscission peaked at two stages (3 and 7 WAA), with the abscission rate plateauing at 10 WAA, by which time 90.09% of the fruit had abscised. By 23 WAA, the fruit had reached the ripening stage and initiated natural abscission. The expression levels of *MiPGs* in the fruit AZ were quantified using quantitative real-time PCR (qRT-PCR) at 3, 7, 10, 16, and 23 WAA, then normalized to the [0,1] interval through min–max scaling across genes and time points for comparative visualization (Figure 4). *MiPG11*, -*12*, -*13*, -*18*, -*20*, -*21*, -*22*, -*23*, -*31*, -*34*, -*41*, and -*51* were not detected or were expressed at low levels, whereas *MiPG7*, -*9*, -*15*, -*33*, -*35*, -*37*, -*52*, and -*53* were highly expressed, especially *MiPG37*, which was the most highly expressed. The expression levels of *MiPG9*, *MiPG37* and *MiPG53* at the peak of fruit abscission (3 and 5 WAA) were higher than those at 10 and 16 WAA. Tissue expression analysis of *MiPG9*, *MiPG37*, and *MiPG53* in flowers, leaves, roots, seeds, and stems revealed that *MiPG9* and *MiPG37* were highly expressed in the leaves and seeds, but *MiPG53* was only highly expressed in the seeds (Appendix A).

The girdling with defoliation (GPD) and ethephon (ET) treatments were applied to macadamia fruit at 5 and 24 WAA, respectively. Both treatments accelerated fruit abscission (Appendix A). Three days after GPD treatment, the cumulative fruit abscission rate (CFAR) was significantly higher than that of the Control (Ctrl) group. Six days after GPD treatment, 93.93% of the fruit underwent abscission, while the CFAR of the Ctrl group was only 16.47%. Similarly, after day 3, ET treatment triggered a rapid increase in CFAR, reaching 21.09% by day 6, compared with 9.59% in the Ctrl group.

As shown in Figure 5, after GPD treatment, the *MiPG9* expression level was significantly greater than that of the Ctrl group on day 1. The *MiPG37* expression level was significantly greater than that of the Ctrl group on days 1, 2, and 3 after GPD treatment; and the *MiPG53* expression level was significantly greater than that of the Ctrl group on days 2, 3, and 5 after GPD treatment. After ET treatment, the *MiPG9* expression level was significantly greater than that of the Ctrl group on days 3 and 5, and the *MiPG37* expression level was significantly greater than that of the Ctrl group on days 1, 3, and 5. The *MiPG53* expression level was significantly greater than that of the Ctrl group on days 1 and 3 after ET treatment. These findings indicate that *MiPG9*, -*37*, and -*53* are involved in macadamia fruit abscission, whereas *MiPG37* may play a more dominant role.

### 2.6. Transient MiPG37 Overexpression Promoted Abscission in Lily Petals

The pCAMBIA3300-*MiPG37* vector was constructed for transient overexpression in lily petals to explore the function of *MiPG37*. As shown in Figure 6, transient overexpression of *MiPG37* resulted in a **78.33%** petal abscission rate in lily flowers at 6 days after infection treatment, whereas the petal abscission rate in the Ctrl group was only 31.21%. These findings indicate that *MiPG37* accelerates petal abscission.

## 3. Discussion

Given that PG serves as a key hydrolytic enzyme mediating pectin degradation through cleavage of α-1,4-glycosidic bonds between D-galacturonic acid residues in polygalacturonan [3,4,28,44], elucidating the functional contributions of *PG* genes to macadamia’s severe physiological fruit abscission becomes imperative. In this study, a systematic bioinformatics analysis of *PG* genes in macadamia was conducted. A total of 56 MiPGs were identified in the macadamia genome and were unevenly distributed on the chromosomes (Appendix A). The number of *PG* genes varies among plant species, such as 38 in citrus [21], 55 in maize [22], and more than 100 in soybean [45], which may be related to differences in the genome size and complexity among species. The majority of *MiPGs* were predicted to be localized only in the cell membrane (Table 1), suggesting that they are secretory proteins involved in cell wall degradation.

Phylogenetic analysis revealed that *MiPG* genes clustered into seven clades (Figure 3), which is consistent with the findings of previous studies [22,26,46]. Consistent with previous reports on *PG* genes in peach [23] and plum [26], the majority of *MiPGs* contained four conserved domains (Table 1). There are four conserved domains in plant PG proteins, and the core amino acid sequences of domains I and II are “SPNTDG” and “GDDC”, respectively. The three aspartic acids (D) in domains I and II may be components of the catalytic sites [47]. Domain III is composed of “CGPGHG”, of which the histidine residue (H) is thought to be involved in the catalytic reaction [48]. The amino acid sequence of domain IV is “RIK”, which may be related to ion interactions at the carboxyl ends of substrates [48]. Domain III is relatively less conserved, which may explain the absence of domain III in as many as the 14 *MiPGs*, except for the homolog closest to *AtQRT3*. Consistent with previous studies, *MiPGs* lacking structural domain III are in clade E (Table 1 and Figure 3) [22]. In addition, some *MiPG* members lack structural domains I, II, and IV, suggesting that they lack catalytic activity or the ability to interact with substrates containing the ionic groups of carboxylic acid groups. Compared with those of other *MiPGs*, *MiPG22*, -*27*, -*28*, -*29*, and -*30* of clade G differed significantly in the absence of any conserved PG domain (Figure 2). In *Arabidopsis*, despite the absence of the PG domain, *AtQRT3* has been shown to degrade the cell walls of pollen mother cells during microspore development [49]. *AtQRT3* is highly homologous to *MiPG-22*, -*27*, -*28*, -*29*, and -*30* (Figure 1), suggesting that it may also have cell wall modification functions similar to those of *PG* genes. Some *PG* genes known to be involved in fruit development, especially abscission, were added to the phylogenetic tree (Figure 1) to identify potential candidate abscission-related *PG* genes in macadamia fruit. Clade C contained the most *PG* genes from horticultural plants, including *TAPG1*, -*2*, -*4*, and -*5* from tomato [17] and *EgPG4* from oil palm [12]. Compared to other evolutionary clades, clades D and E contained more *MiPG* genes. Members of the same clade of *MiPGs* had similar gene structures and conserved domains (Figure 2), similar to the results reported for other plant *PG* gene families [22,26,46]. Thus, the differences in the conserved structural domains and gene structures of *MiPGs* between clades may indicate differences in the composition of pectin that they degrade.

In our study, *cis*-elements were identified and analyzed in the *MiPG* promoter sequences (Figure 3). Numerous light-responsive elements were found in the *MiPG* promoters as well as in the promoters of *Brassica oleracea* [50] and maize [22] *PG* gene family members. Similar findings in the promoters of other cell wall hydrolase genes indicate that cell wall hydrolases are involved in cell wall remodeling during plant photomorphogenesis [51,52]. The promoters of 56 *MiPGs* contained 37 hormone-related *cis*-elements, including ABA, auxin, gibberellin, methyl jasmonate, salicylic acid, and zein (Figure 3). ABA has been extensively shown to promote plant organ abscission [53,54,55], which may explain the high number of ABA-responsive elements contained in the *MiPG* promoters. In addition, some studies have characterized the relationships between *PG* genes and other hormones. In *Arabidopsis*, jasmonic acid regulates floral organ abscission by promoting *QRT2* expression [20]. Gibberellin application may inhibit the abscission of blue honeysuckle fruit by decreasing the expression of cell wall hydrolases, such as PG, cellulase, and pectin methylesterase. Therefore, hormones may be involved in plant development by regulating *PG* expression. However, relevant research is still lacking, especially concerning plant organ abscission. These findings suggest that hormones are also important for macadamia growth and development. However, *PG* function regulation by hormone signaling needs to be further clarified.

Fruit abscission is a complex physiological and biochemical process influenced by multiple cell wall-modifying enzymes [2,3,4]. The transcription profile of related genes during fruit development and ripening can provide important insight for understanding their functions. During macadamia fruit development, abscission peaked at 3 and 7 WAA (Appendix A). By 23 WAA, the fruit reached the ripening stage and initiated natural abscission. Twelve *MiPGs* were not detected or were expressed at low levels in the fruit AZ, whereas *MiPG7*, -*9*, -*15*, -*33*, -*35*, -*37*, -*52*, and -*53* were highly expressed (Figure 4). Although they were not specifically expressed in the fruit AZ, the expression levels of *MiPG9*, *MiPG37*, and *MiPG53* were significantly higher in seeds than in other tissues (Appendix A), suggesting that they may be involved in fruit development. Subsequent studies showed that the expression levels of *MiPG9*, *MiPG37*, and *MiPG53* were significantly higher during macadamia fruit abscission under GPD and ET treatments than in the Ctrl group (Figure 5). These results provide strong evidence that *MiPG9*, -*37*, and -*53* are involved in macadamia fruit abscission. Notably, a member of the E clade, *LcPG1*, in litchi (Figure 1) may play an important role in fruit abscission [30]. *MiPG37* is also located in clade E, suggesting that they may be functionally similar. *MiPG37* expression at 2, 7, and 23 WAA was unexpectedly greater than that of the other *MiPGs*; thus, we constructed a plant *MiPG37* overexpression vector for functional analysis.

The functional validation of genes associated with fruit abscission in most woody plants remains challenging due to the difficulty in establishing genetic transformation systems. *Arabidopsis* and tomato are often used as model plants to study the functions of abscission-related genes [56,57,58,59], but this method is time-consuming and often requires microscopic observation of organ abscission. Therefore, we verified the functions of *PG* genes quickly and directly by transient overexpression in lily petals. Compared with the control, transient *MiPG37* overexpression resulted in premature abscission of lily petals; at 6 days after infection, 78.33% of the lily petals were abscised, whereas 31.21% were abscised in the control (Figure 6). In conclusion, *MiPG37* is closely related to macadamia fruit abscission.

## 4. Materials and Methods

### 4.1. Plant Materials and Treatment

Test trees (*M*. *integrifolia* × *M*. *tetraphylla* cv. HAES 695, Beaumont) were planted in a commercial plantation in Chongzuo, China. At 2 WAA, no fewer than 30 infructescences were labeled on each tree, and the number of fruits on the labeled infructescences was recorded. Fruit abscission dynamics were recorded continuously, and fruit AZs were sampled. A total of four trees were investigated. At 7 WAA, samples were collected from the flowers, leaves, roots, seeds, and stems.

Nine trees were selected and divided into three biological replicates of three trees each. Prior to the physiological fruit abscission peak phase (5 WAA; as determined by previous research [38] and our experimental results, shown in Appendix A), 12 fruit-bearing shoots with a similar number of fruit were labeled at different positions on each tree. For GPD treatment, six shoots were subjected to girdling (a ring of bark approximately 0.8 cm in width, the outer bark, and phloem tissues were removed from the base of the branch) and defoliation (all leaves above the ring were removed). The non-treated shoots served as the Ctrl group. The daily dynamics of fruit abscission were recorded in two shoots per tree, while the remaining shoots were sampled for fruit AZs.

Six trees were selected as test materials, and each tree was considered one biological replicate. Prior to the mature fruit abscission phase (24 WAA; as determined by previous research [60] and our preliminary observation), 12 fruit-bearing shoots with a similar number of fruit were labeled at different positions on each tree. The fruit on the labeled shoots of 3 trees was treated with 2.5 g·L^−1^ ET solution, and water was applied to the remaining 3 trees, which served as the Ctrl. The ET and Ctrl treatment group solutions both contained 0.1% Tween-20. The daily dynamics of fruit abscission were recorded on 2 shoots per tree, and the remaining shoots were sampled for fruit AZs.

In the present study, the cultivar *Lilium* cv. ‘Star Gazer’ was used as the experimental plant material and purchased from the local market in Nanning.

### 4.2. Determination of Fruit Abscission

The CFAR was calculated as a percentage of the cumulative number of abscised fruit divided by the number of initial fruit. The relative fruit abscission rate was calculated as a percentage of the number of abscised fruit on the recorded day divided by the number of remaining fruit in the last record.

### 4.3. Identification of Macadamia PG Gene Family Members

The hidden Markov model profile of the *PG* domain (accession no. PF00295) was retrieved from the Pfam database (http://pfam.sanger.ac.uk/, accessed on 5 March 2022) [26,61], and the PG protein sequences from the *Arabidopsis* genome were downloaded from the *Arabidopsis* Information Resource (http://www.arabidopsis.org/, accessed on 5 March 2022) [46,62]. These domains and the identified *Arabidopsis PG* gene sequences were used as queries against the macadamia genome database (http://macadamiaggd.net/, accessed on 5 March 2022) to perform BLASTP [63]. The selected genome was *Macadamia integrifolia* HAES 741 [40]. All the candidate macadamia *PG* gene family members were subsequently submitted to the CDD (https://www.ncbi.nlm.nih.gov/Structure/cdd/wrpsb.cgi, accessed on 5 March 2022) [64] and the Pfam databases [61] to determine the presence of the domain. The *PGs* encoded by the homologous genes of the well-known *AtQRT3* were also analyzed [49], and five corresponding *MiPG*s were identified.

### 4.4. Multiple Sequence Alignment, Phylogenetic Analysis, and Exon/Intron Structure

Multiple sequence alignment was performed, and conserved domains were analyzed using DNAMAN 6.0 with the default settings. Phylogenetic trees were constructed using the neighbor-joining method with 1000 bootstrap replicates in MEGA 7.0 software (http://www.megasoftware.net, accessed on 6 March 2022) [65], and the output was visualized using Chiplot (https://www.chiplot.online/, accessed on 6 March 2022). The structures of the exons and introns of the *MiPGs* were analyzed using TBtools-II v2.225 [66]. The physiological and biochemical parameters of the full-length proteins were calculated using the ProtParam tool (http://web.expasy.org/protparam/, accessed on 7 March 2022) [26]. The signal peptide and subcellular localization were analyzed using SignalP 4.1 (http://www.cbs.dtu.dk/services/SignalP/, accessed on 7 March 2022) [67] and the CELLO v2.5 server (http://cello.life.nctu.edu.tw/, accessed on 7 March 2022) [68], respectively. MEME motif analysis was performed using MEME (https://meme-suite.org/meme, accessed on 8 March 2022) to identify conserved motifs in the *PG* genes [69], and the maximum number of motifs to be identified was set to 8. The identified motifs were annotated using the Batch Web CD-Search Tool (https://www.ncbi.nlm.nih.gov/Structure/bwrpsb/bwrpsb.cgi, accessed on 8 March 2022) [70].

### 4.5. Cis-Element Analysis of Macadamia PG Gene Promoters

The *MiPG* promoter sequences (2-kb upstream of the start codon) were retrieved from the macadamia genome database (http://macadamiaggd.net/, accessed on 9 March 2022) [63]. The online software PlantCARE (http://bioinformatics.psb.ugent.be/webtools/plantcare/html/, accessed on 9 March 2022) was used to analyze the *cis*-elements in the isolated promoter sequences [43].

### 4.6. Quantitative Real-Time PCR Analysis

Total RNA was extracted using a Plant RNA Kit (OMEGA, Norcross, GA, USA), and cDNA was synthesized using HiScript III All-in-one RT SuperMix Perfect (Vazyme, Nanjing, China) for qRT-PCR. *MiMADH* and *MiGAPDH* were used as reference genes [71]. The reaction was composed of 10 μL of 2×SYBR Green qPCR mix (Biosharp, Hefei, China), 0.4 μL of each primer, 1 μL of cDNA, and 8.2 μL of ddH_2_O, for a final volume of 20 μL. qPCR was performed on a LightCycler 96 (Roche Diagnostics, Mannheim, Germany) using the following program: 95 °C for 30 s; 45 cycles of 95 °C for 10 s, annealing at 60 °C for 15 s, and extension at 72 °C for 15 s. The 2^−△△CT^ method [72] was used to calculate the relative expression of *MiPGs*, and each gene was analyzed three times. The primer information is shown in Appendix A.

### 4.7. Transient MiPG37 Overexpression in Lily Petals

The full-length coding sequence of *MiPG37* (NCBI Reference Sequence: XM_042620705.1) was amplified using specific primers (Appendix A) with 15-bp homologous sequences around restriction sites in a pCAMBIA3300-derived plant expression vector (https://cambia.org/welcome-to-cambialabs/cambialabs-projects/, accessed on 20 December 2022). Recombinant plasmid pCAMBIA3300-*MiPG37* was constructed using the ClonExpress II One Step Cloning Kit (Vazyme, Nanjing, China), which ligates the linearized vector and the target gene fragments. The resulting plasmid construct pCAMBIA3300-*MiPG37* was transformed into *Agrobacterium tumefaciens* EHA105 for further infection.

An improved transient overexpression method for lily petals was developed based on previous studies [73,74,75]. The bacterial mixture was rapidly propagated in a YEB (yeast extract beef) liquid medium containing rifampicin (50 µg/mL) and kanamycin (50 µg/mL) and shaken at 28 °C and 200 rpm until an OD_600_ of 0.8–1.0. The bacterial mixture was subsequently centrifuged at 5000 rpm for 10 min in a refrigerated centrifuge. After centrifugation, the bacteria were resuspended in an infection solution (0.5% (*w*/*v*) phosphate-buffered saline + 0.1 mM acetylsyringone) at an OD_600_ of 0.8. The *Agrobacterium* culture was incubated in the dark for 2 h. Cut lily flowers were selected, and the petals were infected on the first day of bloom. After gently piercing the bottom of each petal with a syringe needle, a mixture of *Agrobacterium* cultures was injected into the petals with 2 mL of bacterial suspension per petal up to the petal AZ. After injection, the lily petals were incubated in the dark for 24 h and transferred to a growth room with light. The *Agrobacterium* culture carrying the empty pCAMBIA3300 vector was used as the negative control. Transient expression assays were performed with 3 biological replicates, and there were 10 flowers per replicate. The petal abscission rate was measured at 6 days after infection, with complete abscission of all petals in an individual lily flower established as the statistical criterion.

### 4.8. Statistical Analysis

Statistical analysis of the data was conducted using SPSS 23.0. Data are presented as the mean ± standard error. The significance of the differences between the treatments was tested using a t-test (*p* < 0.05). All figures were generated using GraphPad Prism 9 and ChiPlot (https://www.chiplot.online/, accessed on 5 May 2023).

## 5. Conclusions

Genome-wide analyses identified 56 members of the *PG* gene family in macadamia, each varying in chromosomal location, gene structure, and motifs, and they were clustered into seven clades. These *MiPGs* consisted of 3–11 exons and 2–10 introns, with the majority containing conserved domains I–IV. The promoters of *MiPGs* contained numerous light-, phytohormone-, and stress-responsive elements. During macadamia fruit development, twelve *MiPGs* in the fruit AZ were either not expressed or expressed at very low levels, but eight exhibited high expression levels. The expression levels of *MiPG9*, *MiPG37*, and *MiPG53* significantly increased during fruit abscission induced by GPD and ET in macadamia. Furthermore, transient *MiPG37* overexpression in lily petals demonstrated that *MiPG37* may play an important role in macadamia fruit abscission.

## Figures and Tables

**Figure 1 plants-14-01610-f001:**
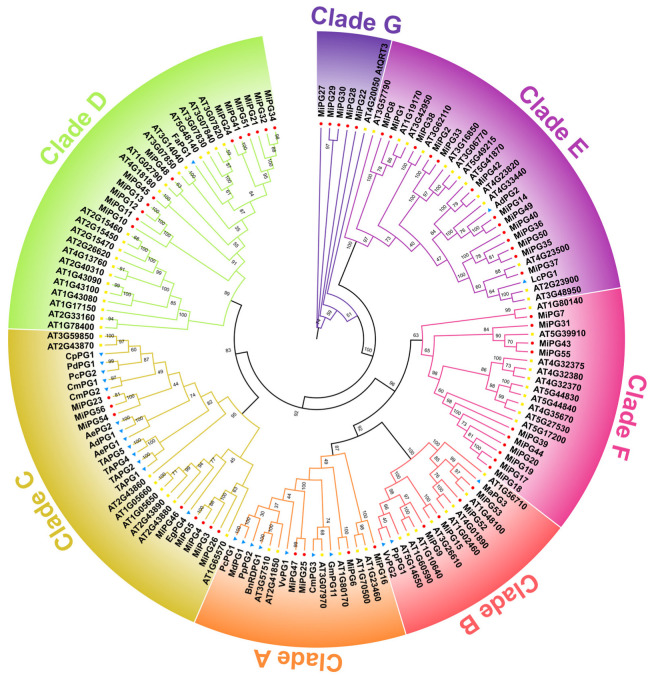
Phylogenetic tree of *PG* genes from macadamia and other horticultural plants. Red circles, yellow squares, and blue triangles represent *MiPGs*, *AtPGs*, and *PGs* from other species, respectively. Different colors represent distinct clades. The values on the branches indicate the bootstrap percentage values for 1000 repetitions. Values less than 50 are hidden. The protein sequences of *PG* genes from other horticultural plants were downloaded from the GenBank database. The sequence information is as follows: apple *MdPG1* (AAA74452), banana *MaPG3* (AY603339), bullace *PdPG1* (DQ375247), grape *VvPG1* (AY043233), and *VvPG2* (EU078975), kiwifruit *AdPG1* (AYP70925), *AdPG2* (AYP70310), *AePGC1* (ARA90624) and *AePGC2* (ARA90625), litchi *LcPG1* (AFW04075), melon *CmPG1* (AF062465), *CmPG2* (AF062466) and *CmPG3* (AAC26512), oil palm *EgPG4* (AFO53698), oilseed rape *BnRDPG1* (Q42399), papaya *CpPG1* (FJ007644), peach *PpPG1* (BAH56488) and *PpPG2* (CAA54448), pear *PcPG1* (BAC22688) and *PcPG2* (BAC22689), soybean *GmPG11* (ABC70314), strawberry *FaPG1* (ABE77145), tomato *TAPG1* (AAC28903), *TAPG2* (AAC28904), *TAPG4* (AAC28905), and *TAPG5* (AAC28906).

**Figure 2 plants-14-01610-f002:**
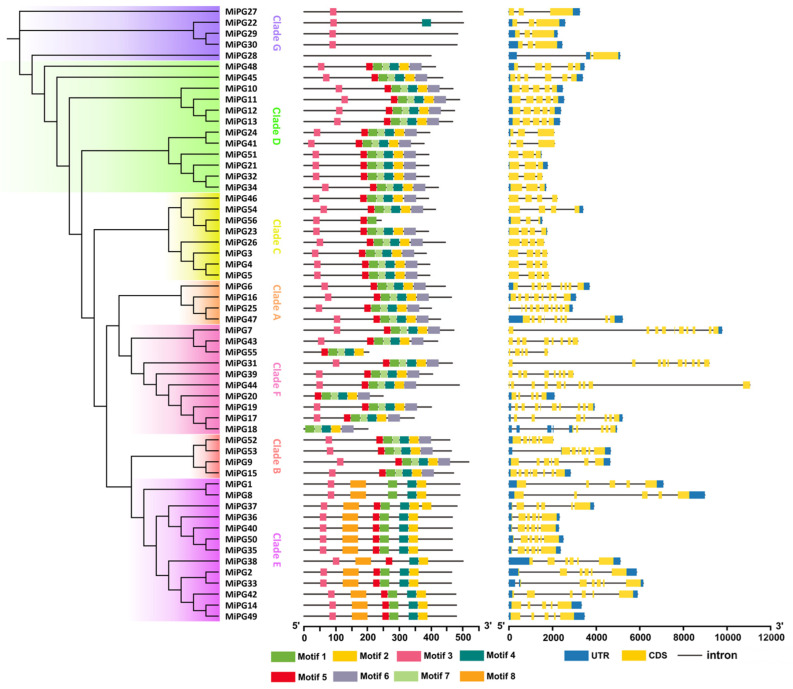
Phylogenetic relationships, motif distribution, and exon–intron structures of *MiPGs*. The left part shows the phylogenetic tree of *MiPGs*, and different colors represent distinct clades. The middle part shows the composition and position of the conserved motifs of *MiPGs*. The right part shows the intron/exon organization of *MiPGs*. UTR (untranslated region), CDS (coding sequence).

**Figure 3 plants-14-01610-f003:**
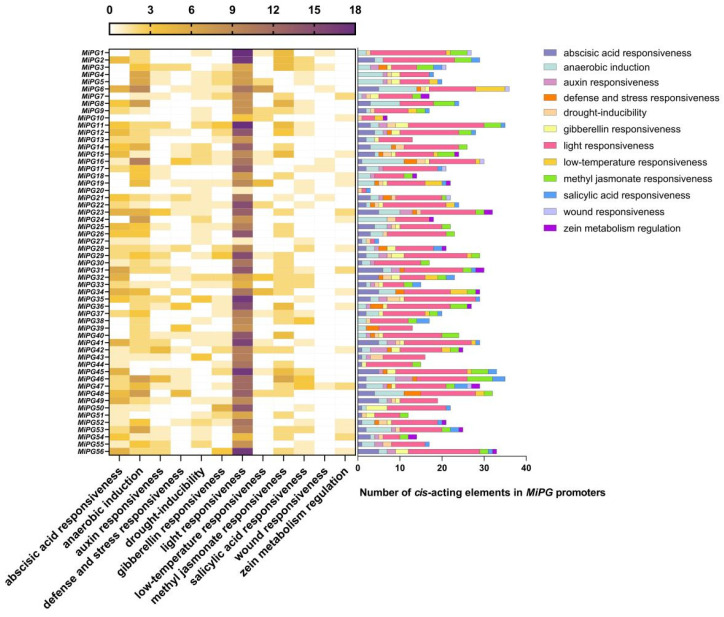
*Cis*-element analysis of *MiPG* promoters. The bar chart on the right shows the number of *cis*-acting elements in the *MiPG* promoters.

**Figure 4 plants-14-01610-f004:**
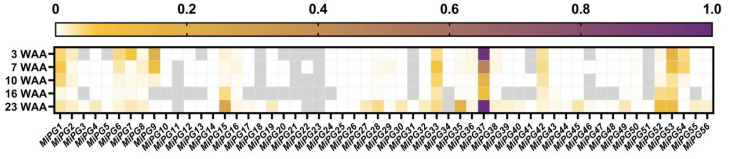
Heatmap of *MiPG* expression levels in the AZ of macadamia fruit at different times after anthesis. Expression levels were normalized to the [0,1] interval using min–max scaling applied across all genes and time points, for comparative visualization. Gray shading indicates undetectable expression. WAA, weeks after anthesis.

**Figure 5 plants-14-01610-f005:**
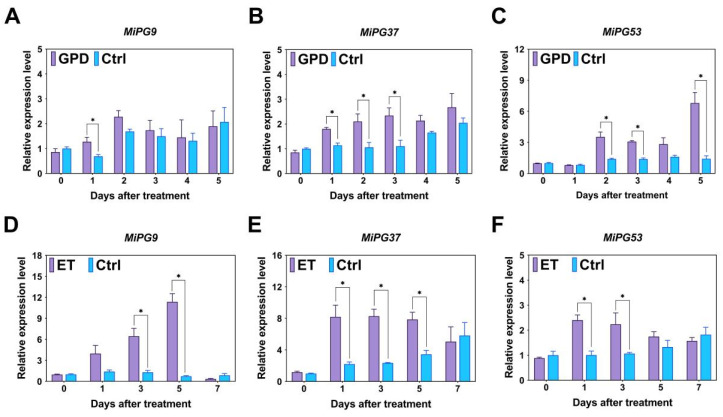
Expression dynamics of *MiPG9*, *MiPG37*, and *MiPG53* under GPD and ET treatments. (**A**–**C**) show the expression profiles of *MiPG9*, *MiPG37*, and *MiPG53*, respectively, under girdling with defoliation (GPD) treatment. (**D**–**F**) show the expression profiles of *MiPG9*, *MiPG37*, and *MiPG53*, respectively, under ethephon (ET) treatment. Significant differences at the 0.05 level according to the *t*-test are indicated with asterisks (*).

**Figure 6 plants-14-01610-f006:**
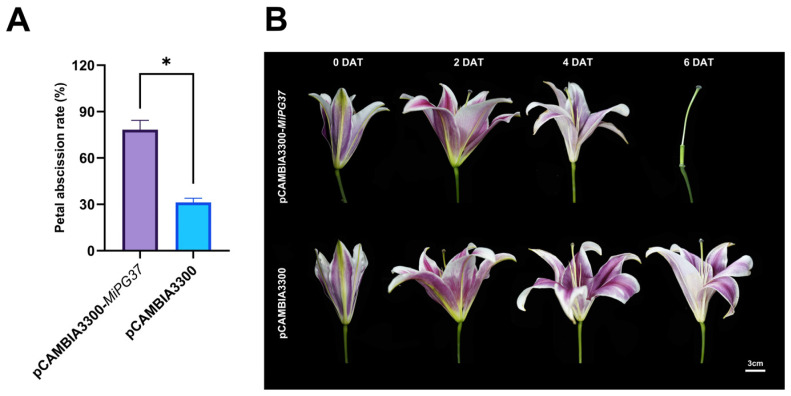
Effects of *MiPG37* overexpression on lily petal abscission. (**A**) Petal abscission rate at 6 days after treatment. (**B**) Phenotypic progression of petal abscission on different days after treatment (DAT). The empty pCAMBIA3300-derived vector was used as the negative control. Significant differences at the 0.05 level according to the *t*-test are indicated with asterisks (*).

**Table 1 plants-14-01610-t001:** Basic information on *MiPGs*.

Gene Name	Gene ID	Deduced Protein			Signal Peptide	Subcellular Localization	Domain
		Length (aa)	Molecular Weight (kDa)	Isoelectric Points (pI)			
MiPG1	LOC122081278	491	55.10	8.76	−	CM	II IV
MiPG2	LOC122081682	465	50.10	5.10	+	CM	I II IV
MiPG3	LOC122063767	385	41.23	9.25	+	CM	I II III IV
MiPG4	LOC122063777	396	42.08	7.93	+	CM	I II III IV
MiPG5	LOC122063785	396	42.09	8.66	+	CM	I II III IV
MiPG6	LOC122071976	445	48.72	8.34	+	CM	I II III IV
MiPG7	LOC122064844	472	50.83	8.94	−	CM	I II III IV
MiPG8	LOC122066229	491	54.99	8.59	−	CM	II IV
MiPG9	LOC122073447	519	56.38	7.52	+	CM	I II III IV
MiPG10	LOC122074598	469	48.01	7.47	+	CM	I II III IV
MiPG11	LOC122074599	490	49.83	8.14	+	CM	I II III IV
MiPG12	LOC122074600	474	48.98	8.28	+	CM	I II III IV
MiPG13	LOC122074601	468	48.34	7.47	+	CM	I II III IV
MiPG14	LOC122072816	480	52.49	6.79	−	CM	I II IV
MiPG15	LOC122075419	471	51.08	8.77	+	CM	I II III IV
MiPG16	LOC122076188	464	50.78	4.85	+	CM	I II III IV
MiPG17	LOC122077755	347	37.08	8.88	+	CM	I II III IV
MiPG18	LOC122078590	201	21.08	4.85	−	CM	II III IV
MiPG19	LOC122078589	401	42.42	4.94	+	CM	I II III IV
MiPG20	LOC122078252	249	26.32	6.14	−	CM	I II III IV
MiPG21	LOC122082848	393	42.78	8.91	+	CM	I II III IV
MiPG22 *	LOC122081461	502	54.13	5.56	+	CM Chl Cyt	
MiPG23	LOC122083126	392	41.78	5.88	+	CM	I II III IV
MiPG24	LOC122084718	396	42.92	9.42	+	CM	I II III IV
MiPG25	LOC122089024	401	42.97	5.80	−	CM	I II III IV
MiPG26	LOC122092380	445	47.66	5.99	+	CM	I II III IV
MiPG27 *	LOC122092659	498	54.13	9.45	+	Chl	
MiPG28 *	LOC122093358	400	44.51	5.34	−	Chl	
MiPG29 *	LOC122094310	484	51.71	8.26	−	CM	
MiPG30 *	LOC122094116	482	51.89	8.55	+	CM Chl	
MiPG31	LOC122093962	467	50.92	5.33	−	CM	I II III IV
MiPG32	LOC122092887	394	42.77	8.86	+	CM	I II III IV
MiPG33	LOC122093810	464	49.69	5.23	+	CM	I II IV
MiPG34	LOC122057274	423	46.27	8.59	−	CM	I II III IV
MiPG35	LOC122058248	467	50.76	6.08	+	CM	I II IV
MiPG36	LOC122057404	467	50.67	6.70	+	CM	I II IV
MiPG37	LOC122058173	482	52.29	6.32	+	CM	I II IV
MiPG38	LOC122058047	501	55.45	6.46	−	CM	II IV
MiPG39	LOC122059778	405	43.35	6.06	+	CM	I II III IV
MiPG40	LOC122059401	467	50.60	6.31	+	CM	I II IV
MiPG41	LOC122059027	378	40.99	9.16	−	CM	I II III IV
MiPG42	LOC122061528	478	51.97	8.30	−	CM	I II IV
MiPG43	LOC122061700	421	46.05	5.41	+	CM	I II III IV
MiPG44	LOC122061732	489	51.74	5.61	+	CM	I II III IV
MiPG45	LOC122062845	437	45.85	8.82	+	CM	I II III IV
MiPG46	LOC122065217	392	41.54	9.43	+	CM	I II III IV
MiPG47	LOC122065260	430	47.16	5.18	+	CM	I II III IV
MiPG48	LOC122065662	414	44.01	8.75	+	CM	I II III IV
MiPG49	LOC122066012	480	52.51	7.10	−	CM	I II IV
MiPG50	LOC122066415	467	50.62	6.01	+	CM	I II IV
MiPG51	LOC122066705	393	42.81	5.54	+	CM	I II III IV
MiPG52	LOC122068767	459	50.30	5.98	+	CM	I II III IV
MiPG53	LOC122068773	464	49.70	4.99	+	CM	I II III IV
MiPG54	LOC122069182	414	43.73	8.45	−	CM	I II III IV
MiPG55	LOC122070809	204	21.71	6.21	−	CM	I II III IV
MiPG56	LOC122071794	243	26.05	9.08	+	CM	I II

* denotes the *MiPG* orthologs of Arabidopsis *QRT3.* Subcellular localization predictions: CM, cell membrane; Chl, chloroplast; Cyt, cytoplasm; +, present; and −, absent.

## Data Availability

The original contributions presented in this study are included in the article/Appendix A. Further inquiries can be directed at the corresponding author.

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
