# Peer review of "Genome-Wide Analysis of the Polygalacturonase Gene Family in Macadamia and Identification of Members Involved in Fruit Abscission"

_plants, 2025, doi:10.3390/plants14111610_

Round 1

Reviewer 1 Report

Comments and Suggestions for Authors

The authors analyzed the protein sequences of putative Polygalacturonases from a Macedemia genome database. The protein sequences were analyzed and compared to polygalacturonases from other plants, and the expression levels of Macadamia putative polygalacturonases were measured at different stages and under different conditions. Heterologous expression of one selected polygalacturonase was performed.

Interesting work, I have some suggestions to improve the manuscript:

First, it is not clear which species or which of the available genomes were used. Currently, there are two options for Macademia integrifolia, please make clear the database used. 

Table 1 needs an improved description, and to improve understanding, I suggest that the conserved domains could be described here rather than in the M&M section.

Figure 1 needs to show that the AGI: At4g20050 is equivalent to AtQRT3. This correlation is not explicit in the text or the tree.

In Figure 3, yellow is considered "less", but there are different shades of yellow, same with purple, which is described as "more". In the figure already exist a clear color scale. I suggest describing or just keeping the color scale. 

Figure 4. As Figure 3, there exists a clear color scale. The figure seems to use the expression levels of MiPG37 (the highest expression level) to normalize the data. Please make clear how the normalization was done.

Reviewer 2 Report

Comments and Suggestions for Authors

The manuscript “Genome-Wide Analysis of the Polygalacturonase Gene Family in Macadamia and Identification of Members Involved in Fruit Abscission” by Fei et al presents an extensive analysis of the Poligalacturonase gene family that include the identification of 56 members PGs in the macadamia genome, phylogenetic and bioinformatic analysis and expression profiles.

This manuscript is well written and contains enough information related to the relevance of the study of the Macadamia PGs family to understand their role in fruit abscission to finally increase the macadamia yield.  The results are clearly presented and discussed.

In general, the Material and Methods section clearly described the experimental design.  However, it is not clearly described why the GDP and ET treatments were done at 5 and 24 WAA.  An image of the lily petals that were transiently transformed could improve figure 6.

The discussion must be rewritten because there is information that is already present in the introduction.

Minor comments

Lane 52: Pectin methyl esterases are noy Hydrolases

Lane 114: Macadamia integrifolia must be written in cursives

Table 1: subcellular locationa must be written in Subcellular localization
